# Leveraging LLMs to Improve Hardware-Software Co-Design Workflow Productivity and Accessibility

Kavya Sreedhar[1], Josh Ogbonda[1], Pengqi Yin[1], Narges Shahidi[1], Kanthi Nagaraj[1], Zhijie Deng[1], Rami Cohen[1],
Ton Kalker[1], Sameer Kumar[1], Amir Yazdanbakhsh[2], Suvinay Subramanian[1]
[1]Google [2]GoogleDeepMind
kavyasreedhar@google.com

*Abstract*—Hardware-software co-design workflows are critical for chip design. However, these workflows are often manual and require expert knowledge. In this paper, we propose leveraging LLMs to improve the productivity and the accessibility of these workflows. With customized prompt engineering, we can enable LLMs to generate architecture insights from various large data sources describing the performance of models on different hardware systems. We create a taxonomy for the stages of questions that users ask during a typical co-design workflow. This taxonomy provides a way to logically reason about performance data and evaluate the capabilities of LLM-integrated workflows. We then introduce our prototype system, VIEW, which relies on custom prompting with Gemini 2.0 Flash. VIEW is currently able to replace humans in the loop for some types of questions, improving productivity, and explain its reasoning when arriving at conclusions, improving accessibility. To illustrate the importance of our prompt engineering, we evaluate Gemini 2.0 Flash out of the box, without any prompting, in this workflow. Unlike VIEW, Gemini 2.0 Flash by itself hallucinates answers when data is not available. We hope this preliminary proof of concept encourages the community to further pursue this direction of research.

## I. Introduction

Hardware-software co-design is critical for designing efficient accelerators for deep learning models [15]. The increasing scale and complexity of Large Language Models (LLMs) [1], [2], [14], [25], coupled with the duality of the rapidly-evolving application landscape and the naturally-slower pace of hardware development and fabrication, make it difficult to quickly iterate in co-design workflows [19], [33]. Furthermore, a key challenge in realizing the full potential of co-design lies in the vast and complex design space encompassing both algorithmic and architectural choices.

To explore this design space, traditional co-design methodologies often rely on expert knowledge in model developments (e.g., attention [28], flash attention [9], mixture of experts [24], long-context modeling [23], [34]) and specialized hardware (e.g., tailored processing elements [7], [17], multi-level memory hierarchies with complicated dataflows [5], [20], [27], [29]). This workflow can be manual and time-consuming. Some works leverage LLMs in this workflow, but they can require computationally-expensive automated search techniques [3], [6], [12], [21], [35] or retraining [21], [22].

Consider the traditional co-design workflow shown in Figure 1. The user is given some performance data. This data is broadly defined: for example, this data can be a collection of measurements from executing LLMs on real hardware

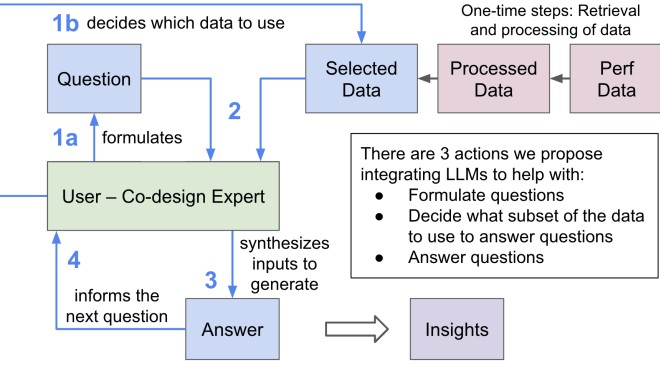

In a traditional **co-design workflow**, users typically go through an iterative Q&A process to generate **insights** from **performance (perf) data**.

Fig. 1: The traditional hardware-software co-design workflow is iterative (the blue loop for steps 1 to 4). Given a large amount of performance data, the user starts by asking questions and deciding what subset of data to look at for answers. The user works to answer these questions, which then help the user formulate following questions until they arrive at a model or architectural insight. We propose integrating LLMs into this workflow, as shown in Figure 2, to make this workflow more accessible, efficient, and automated.

systems, simulation results, or rough spreadsheet models. Furthermore, this data can be aggregated from various sources and provide different views of executing various models on various hardware systems, from detailed performance metrics to high-level summaries. The data input does not need to be limited to raw performance data. Other data sources such as hardware system specifications, model hyperparameters, code describing compiler optimization passes, and analytical equations can provide more context. As a result, this data can be large and may require a one-time pre-processing step.

The user typically iteratively formulates questions (1a) to make sense of this data. Every iteration, the user has to decide what subsets of the data to focus on to ask questions, and what data is relevant to answer these questions (1b). Given the question and the data (2), the user answers the question (3). This process repeats: the user continues asking questions, using what they learn from previous answers (4). With enough answers, the user can determine insights about the evaluation.

In this paper, we propose integrating LLMs themselves into this co-design workflow to improve user productivity and accessibility. In Figure 1, the green box represents the user,

who is typically a co-design expert. We formulate the key tasks the user is responsible for in this flow, with output arrows from the user: formulating questions, selecting what data subset is relevant for the questions being asked, and generating the answers. To replace/aid the user for any combination of these three tasks, we leverage the emergent reasoning and generative capabilities of LLMs. Our work builds upon recent advancements in LLM prompting [4], [8], [10], [26], [30], [31] for our customized prompt engineering. With these prompts, and our pre-processing of raw data sources, we enable LLMs to synthesize information and answer key questions that arise during the co-design process.

We next introduce a taxonomy for the stages of increasingly-complicated questions users ask during a typical co-design flow. This taxonomy also provides the basis for a metric to evaluate the capabilities of LLM-integrated workflows. We believe that building upon this formulation is a critical task for the co-design community: this will allow us to construct an open standardized benchmark to evaluate LLM-integrated co-design workflows.

We first evaluated Gemini 2.0 Flash out of the box to understand the necessity of prompt engineering. While effective for the first stage of questions ("Stage 1"), this baseline approach struggles when data is sparse, often hallucinating incorrect answers or conflating disparate data sources. Our prototype system, Visualization Insights and Exploration Workbench (VIEW), overcomes these limitations by integrating domain-specific knowledge. VIEW leverages our understanding of common co-design workflows to retrieve an accurate subset of simulation data relevant to the user query. It then provides the LLM with a custom prompt that contextualizes how to interpret the selected data. This synergy of targeted retrieval and associated context improves response accuracy and reliability.

Unlike general-purpose retrieval-augmented generation systems relying on vector embeddings [13], VIEW employs a more focused strategy. It further uses another LLM informed by domain awareness to *automate* data selection. VIEW explicitly communicates when an answer cannot be reliably determined—a vital feature for less-seasoned users building intuition. With LLMs in the loop to automate and explain the co-design process, VIEW is a start towards improving productivity, and enabling non-experts to derive meaningful insights. We hope our problem formulation and taxonomy stimulate further research in accelerating the co-design process.

## II. REASONING ABOUT PERFORMANCE DATA

There are different logical sequences of questions that users may have about the performance data input in Figure 1, referred to as "data" below. We provide a taxonomy to classify these types of questions. We consider a high-level grouping of questions, denoted stages, which consist of progressively harder questions to reason about:

- Stage 1 — **What** is the data?
- Stage 2 — **Why** is the data the way it is?

- Stage 3 — **How** can I change the model or the hardware system, to get a different desired result or make it match an expected result?
- Stage 4 — **Where** in the data should I look to focus my analysis? What questions should I ask about the data?

For most Stage 1 questions, the answer is directly available within the performance data. The exceptions are Stage 1 "Data Query +" questions, explained later in this section. For questions in the remaining stages, the LLM has to reason about the raw data given to answer these questions. In other words, these answers are *not* directly present in the performance data.

Stage 1 questions are data queries or data comparisons. These questions do not require co-design, model, or hardware knowledge and can be answered by anyone with the data. Data queries require simple lookups in the data. Examples include "What is the execution time for Q projection for Llama2-70B on a state-of-the-art GPU system?" and "Which collectives were required, and which operators were they after?".

Basic data comparisons require multiple data queries, with answers present in the data. Then, these questions require comparisons. The LLM needs to extrapolate which data queries are required for the comparison and how to compare these numbers. Example questions include "What are the performance bottlenecks?", "Is the Q projection communication-bound or compute-bound?", and "How does the distribution of FLOPs required by model operations correspond to the percentage of total execution time required for these operations? Are some operations not efficiently executed on this hardware?". The last example guides the user on which operations to focus on optimizing. These questions can extend to data comparisons across many different workloads on many hardware systems.

Stage 1 questions can go further than simple lookup and comparison by relying on knowledge that a previously-trained LLM would have. We refer to these questions as "Data Query +" questions, which do require some expert knowledge in order to better contextualize the results. For example, to get a sense of how a current experiment compares to prior standards, we could ask "Can you compare this simulator data of serving this model on a mock hardware system to measurements for serving on a state-of-the-art GPU?". As another example, to evaluate gaps in simulator accuracy, we could ask "Are there any model operators not simulated in these results for this model? What proportion of FLOPs do these missing operators comprise for this model?".

Stage 2 requires reasoning about consequences to explain what causes the data to be the way it is. These questions do require some co-design, model architecture, or hardware system knowledge for a human to answer. Importantly, the answers to these questions are not directly in the performance data, and the LLM must extrapolate to understand causes and effects. Example questions include "Why does the same workload consume less power on hardware system xPU-A than xPU-B?", "Why can the weights for Q projection fit in the lowest level of the memory hierarchy for model A but not for model B?" and "Why do the feedforward layers require the largest proportion of execution time on xPU-A?".

| Label | Stage | Question | Answer |
|-------|-------|----------|--------|
| 1 | 1 | Which system achieves higher performance for this workload? | xPU-A is 15% faster |
| 2 | 1 | Which operator has the highest performance difference between the two systems? | Q projection |
| 3 | 2 | Why is the Q projection runtime different between the two systems? | xPU-A pins all the Q projection weights in local SRAM, while xPU-B has to fetch these weights from off-chip HBM. |
| 4a | 2 | Why can A pin the weights while B cannot? | xPU-A has a 15% larger on-chip SRAM, and the Q projection weights size is in between xPU-A and xPU-B's on chip-SRAM capacity. |
| 5a | 3 | If we increase xPU-B's on-chip SRAM by 7% so the Q projection weights fit, will xPU-A and xPU-B have the same performance? | No, xPU-A also pins the output projection weights while xPU-B does not, and that operator's weights will still not fit in this larger SRAM. |
| 6a | 3 | How much do I need to increase xPU-B's on-chip SRAM by to get the same performance as xPU-A? | A 10% increase in SRAM capacity for xPU-B will produce the same runtime for Q projection and output projection, but there may be other differences in the hardware resulting in performance differences. |
| 4b | 3 | Why can xPU-A pin the weights while xPU-B cannot? xPU-A and xPU-B should have the same on-chip SRAM capacity and HBM capacity. | xPU-A and xPU-B have the same on-chip SRAM and HBM capacities, but it looks like the activations for this operator require 2× the memory in B. |
| 5b | 3 | Why do the activations require more memory in xPU-B? | The data type for the activations in FP8 for xPU-B and INT4 for xPU-A. The workload is not the same. Please run your evaluations again with the same INT4 data type for the workload for both systems. |

TABLE I: Two examples of logical sequences of questions a user may ask about performance data for the execution of a workload on two different hardware systems xPU-A and xPU-B, with expected answers and questions classified by Stage type.

With the answers from Stage 1 and Stage 2 questions, the user better understands the characteristics of the performance results they have. The value addition of bringing an LLM into the loop for Stage 1 questions is to speed up the lookup and comparison process when going through a lot of performance data. These questions can also provide some further context for these specific lookups and comparisons. In Stage 2, an LLM helps the user in two ways: first, in saving time in hunting down different possibilities that may explain the resulting data, and second, in guiding the user in where to look for causal relationships for the workload and architecture. An LLM capable of reasoning about these dependencies, even when the answers are not explicitly available, can explain its insights to an end-user. This capability not only saves time for experts, but it also enables non-experts to actively build intuition in reasoning about performance data.

Stage 3 requires building upon the understanding from Stage 1 and Stage 2 answers to further reason about whether the data makes sense or what would need to be done to make the data look a certain way. For a human, these questions are typically answered by a co-design expert with a deep knowledge of the models and the hardware architectures. We identify two different use cases in this stage.

In the first use case, Stage 3 questions ask if the data matches user intuition. If the data does not match intuition, Stage 3 questions enable going further to discern what features of the workload and/or the system cause the discrepancy. For example, consider a model layer requiring $F_l$ FLOPs and a hardware system with $F_h$ FLOPs. If the evaluation reports FLOPs utilization $U$ and compute execution time $t$, we expect $t = \frac{F_l}{F_h * U}$. If hardware system xPU-A has 2× the FLOPs of xPU-B, where $F_l > F_{h(B)}$, we would expect that $t_{(A)} < t_{(B)}$.

There are two reasons there can be a data/intuition mismatch. In one case, the evaluation is correct, and this discrep-

ancy highlights a misunderstanding in the intuition. Then, the LLM can help the user better understand the flaw in their logic. Alternatively, the intuition is correct, and there is a bug in the evaluation. For example, this bug could be in the evaluation setup, model or system definitions, or underlying implementations. If there is a bug, the LLM-integrated workflow can help pinpoint the issue, and suggest an approach for the fix.

In the second use case for Stage 3, the user understands why the data looks the way it does, and would like to understand what to change in the system and the workload to produce some other behavior. These questions can be more architecture-focused, e.g., "How can I minimize memory transfers for the Q projection operator in this model on this system?", or more model-focused, e.g., "How many attention heads should my model have for more efficient mappings to a given hardware architecture?". Note that the LLM would need to have some sense of model quality for the second example, and know not to produce trivial answers such as zero. This understanding could already be present from the LLM's training, or may require fine-tuning, augmentation with further data sources, or more refined custom prompting.

Stage 4 serves as a meta-stage that reasons about what questions should be asked in Stages 1, 2, and 3. Stage 4 questions direct the user on what is interesting in their data, what trends appear, and where to look and focus their understanding or debugging efforts. This stage does not necessarily require specific questions prompted by the user as in the other stages. Instead, Stage 4 allows for reasoning about the takeaways from this data *without* that explicit direction from the user.

Having an LLM reason about this data and answer Stage 3 questions would qualitatively save significant time, thus improving the productivity of this workflow. Experts can now rely on the LLM to sort through various data sources and check the LLM's reasoning with their expectations. Stage 3 answers

also help individuals with less specialized domain knowledge ramp up in this space, improving accessibility. Stage 4 goes further to direct the user in what to look for and takeaway from the data. Importantly, Stage 4 questions move the burden of the question sequence generation from the user to the LLM. This shift opens up this workflow to users who are not concerned about the specifics and only want high-level insights. Overall, this stage automates more of the workflow and enables guiding a user with any level of expertise.

### A. Example Workflows

We show examples of sets of questions that we asked in recent co-design studies in Table I. A user is comparing the performance of a workload on two different hardware systems, xPU-A and xPU-B. For this workload, xPU-A achieves better performance than xPU-B, and the user wants to understand why. User 1 asks questions 1, 2, 3, 4a, 5a, and 6a, while User 2 asks questions 1, 2, 3, 4b, and 5b. For the sequences of questions shown, both users are co-design experts looking to deeply understand the particulars of the data.

User 1 wants to improve xPU-B so that xPU-B achieves similar performance for this workload as xPU-A. Questions 1 and 2 are Stage 1 questions to understand what the data is, while questions 3 and 4a are Stage 2 questions to understand what causes the data to be that way. Finally, questions 5a and 6a are Stage 3 questions to understand what to change in xPU-B for this workload to achieve the same performance as when run on xPU-A. These Stage 3 questions consider the first Stage 3 use case of how to match desired behavior.

A Stage 4 question would allow for bypassing all of these steps. The user could directly ask "Which system achieves higher performance for this workload, and how could I improve the lower-performing system to get similar performance to the better system?". The LLM, not the user, would then automatically follow a similar reasoning breakdown as shown in Table I, with the answer directly showing the question 6a answer of increasing the SRAM capacity by 10% to improve the Q and output projection execution times for the systems to have similar performance. The Stage 4 question opens up this understanding process to non-experts. Importantly, the Stage 4 question flow enables users looking for high-level insights to walk away with this overall understanding of the data without having to delve into the specifics of the data shown in the breakdown of questions, greatly improving accessibility.

In contrast, User 2 exemplifies the second case in Stage 3, where this workflow could assist with finding bugs in the performance data. We see that the logical sequence of questions are the same between User 1 and User 2 up until question 3. Question 4b is a Stage 3 question that dives into discrepancies between the data and the user's expectation. Question 5b is another Stage 3 question, where the answer identifies an inconsistency in the workload evaluation and provides a solution to fix the bug. A Stage 4 question would again enable bypassing all these steps and directly asking "Which system achieves higher performance for this workload and why?". The LLM could then reason about the data to conclude that the

performance difference is due to using different data types in the workload for different systems. The LLM can further suggest using the same data types for fair comparison.

## III. THE VIEW PROTOTYPE SYSTEM

### A. Workflow Setup

In our specific use case, we are synthesizing takeaways from simulator data showing how various deep learning workloads perform on different hardware architectures. Our internal simulator builds from the roofline model [32]. In this workflow setup, a user typically conducts some sweeps, which are Cartesian products of various execution configurations for a workload. These configurations could include various batch sizes, sharding strategies, and sequence lengths, simulated on a set of GPU or TPU systems that are modeled in the simulator.

Evaluating all of these possible combinations can generate a significant amount of data. To narrow the data to look at, we focus on Pareto-optimal [11] execution configurations, when considering the performance normalized by the total cost of ownership (TCO) of the hardware system [16] versus the latency of serving or training. This forms a Pareto curve of execution configurations, with a mock graph shown in the "Data Visualization" in Figure 2. Every hardware system that the workload was evaluated on has its own set of execution configurations on the Pareto frontier, as the example shows for xPU-A and xPU-B. Then, we consider only these Pareto-optimal points and the resulting performance metrics on various hardware systems to limit the amount of data. These metrics include operator-level execution time breakdown, FLOPs utilization, and the location of weights in the memory hierarchy.

In a typical workflow, we would need to manually look through these Pareto-optimal points to find specific areas of interest, such as where two systems may differ in performance for the same workload. In this example, we would then examine the details of the performance metrics for the different hardware systems to determine what factors cause the observed differences. This investigation often results in insights about the workload and the hardware (or identifies bug(s) in the evaluation). This process exemplifies how co-design workflows can require a lot of iterations and also rely on expert knowledge about the features of the workload and target hardware system to know where to look and dig deeper.

### B. VIEW Overview

We built the VIEW prototype system, shown in Figure 2, to aid our day-to-day co-design workflows. A user enters questions in a chat box on the user interface (UI). A question triggers an LLM call to select the subset of data to focus on the answer the question. The Data Retriever and Filter, as the name suggests, gets and filters the chosen subset of data from all of the simulator performance data. This data may be further processed and then input to another LLM call, which generates a response presented back to the user via the UI. Both LLMs are provided different custom prompts to provide context for their specific tasks, as explained in Section III-D.

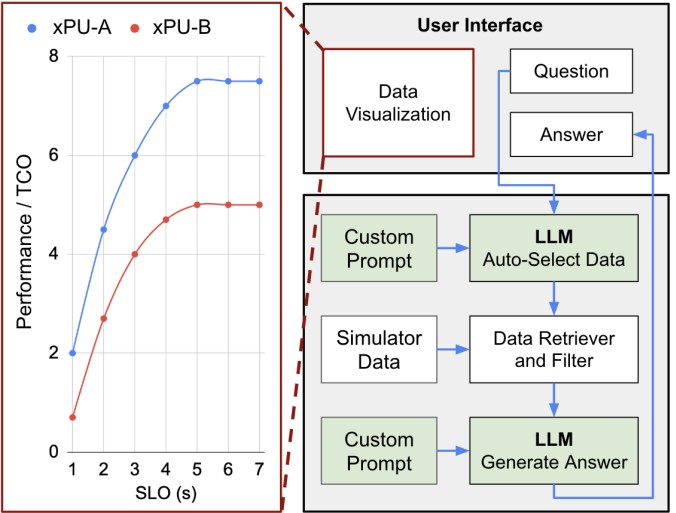

Fig. 2: VIEW system (Workflow 5 in Figure 3). Arrows show the system process. Green boxes indicate LLM integration.

We intentionally construct an intuitive conversational UI. With this UI, we hope to lower the activation barrier for early co-design researchers who may not have specialized knowledge in deep learning workloads and hardware design. The VIEW UI also allows the user to interact with the Pareto Curve graph, with a mock example shown in Figure 2. VIEW also provides other data visualizations for specific execution configurations. These interactive graphs and tables can further help the user formulate questions.

### C. Integrating LLMs into this Workflow

We include an LLM, Gemini 2.0 Flash, with prompting, in two places in this workflow, indicated by green boxes in Figure 2. First, consider the "Auto-Select Data" LLM. To motivate this LLM integration, note that even the files with only the Pareto Curve subset of data from the model sweeps can be hundreds of megabytes. This file size easily exceeds the sequence length for current LLMs, so it is necessary to select what data to focus on when looking to answer questions.

To address this challenge, VIEW integrates an LLM to select what data is required to answer a question. In this case, the LLM is given the question asked by the user. Then, the LLM is tasked with deciding which data subset is relevant to answer this question. This is a nontrivial task since the input data is coming from many different data sources and several files may need to be retrieved. In addition, this selection is not an obvious lookup. Unlike many search problems, there are no key words to guide the selection. In our case, the data consists of many numbers, where there is no clear indication of what data is relevant. Thus, the context provided in the prompting is critical to enable the LLM to reason about what data is required, and which data sources can provide that information.

Currently, VIEW supports asking questions about specific configurations on the Pareto curve, trend and best-of questions across a Pareto curve, and comparisons between multiple system. For example, if a user asks about the performance of a specific execution configuration, VIEW selects only the

data for that configuration. If a user instead asks to compare the performance of a workload between two different systems, VIEW is capable of recognizing that the characteristics and performance metrics for the Pareto-optimal points for *both* systems in the question, not just one system, must be selected.

Second, the "Generate Answer" LLM synthesizes the various data sources selected and answers user questions. Having an LLM answer user questions saves the user the significant time of hunting through data for specific points on the Pareto curve and manually comparing them to arrive at a conclusion. Importantly, the LLM is able to explain its reasoning process to the user. For an expert user, this reasoning serves as a sanity check to see if this data and process are expected. For a less seasoned user, this reasoning equips them to build intuition on where in this vast amount of data to focus on, and how the model and hardware together affect performance.

### D. Custom Prompt Engineering

This section describes the custom prompts we developed to provide the LLMs context. Section V then provides examples of the consequences of specific prompting. For the "Auto-Select Data" LLM, our prompt frames this task as a classification problem: the LLM is asked to classify the user question into one of several question types. The prompt provides at most three-sentence descriptions of each data source available from the simulator, such as summaries of the Pareto-optimal execution configurations and detailed performance metrics for all evaluated configurations (Pareto-optimal or not). The prompt then specifies the possible question categories, and which data sources to pull in depending on the classification. For example, to compare Pareto-optimal execution configurations on several different systems, the prompt would direct the LLM to retrieve several files, one for each system with the relevant performance metrics. The prompt includes two examples of user questions for each category.

The prompt for the "Generate Answer" LLM starts with a general tool-agnostic introduction paragraph that tells the LLM that it is an "ML systems performance expert", describes roofline models, and explains the basic structure of LLM workloads. The rest of the prompt is constructed dynamically based on the question classification determined from the first "Auto-Select Data" LLM. This prompt is then combined with the relevant data selected from the database and the conversation history.

For questions about individual execution points, the prompt explains the formatting of the data (to illustrate, for example, how repeated layers are represented), and the type of performance metrics included. For our simulator data, there are attributes (e.g., FLOPs, tensor shapes, number of bytes read/written from memory, number of bytes communicated across chips for weights) and metrics derived from these attributes and hardware system parameters (e.g., time spent in compute, time spent in memory reads/writes, time spent on collectives, tensorcore power).

For questions about Pareto curve trends or multi-system comparisons, the prompts are very short: these prompts simply

direct the LLM to use metrics like latency and performance per TCO to compare different execution configurations. The multi-system category specifies that these comparisons are between multiple systems. This infrastructure can be extended to compare multiple workloads as well.

These prompts do *not* contain any information about the stages described in Section II, or any further classifications within a stage (e.g., data query vs data comparison vs data query + for Stage 1, or the two use cases in Stage 3). The LLM extrapolates how to answer various questions on its own.

## IV. EVALUATION SETUP

We use VIEW as a proof of concept to demonstrate the value of LLM-integrated co-design workflows. We generated simulator data for various execution configurations of serving Llama2-70B on TPUv5. Using this data, we evaluated various co-design workflows for some question types. For each co-design workflow row listed in Figure 3, we asked five representative questions for each of the three different types of Stage 1 questions, as discussed in Section II: "Data Query", "Data Comparison", and "Data Query +".

"Data Query" questions have answers that are directly available in the CSVs of numbers that are the data input. Note that the column names may not correspond to the terms used in the questions (e.g., column title is *roofline* but the question asks about *execution time*). "Data Comparison" questions require comparing several values available in the data, but neither the question nor the prompt provide any guidance on what numbers to compare or how to do this comparison.

"Data Query +" questions describe data queries that rely on broader knowledge not available in the simulation data; the LLM has to rely on its trained knowledge for answers. An example question from our evaluation is "Are there any model layers missing from these results for Llama2-70B, based on what you know about this model architecture? If there are, please provide a table listing the layers in this model and specify which ones are modeled/not modeled in the results data." In this example, we posit that an LLM would be able to reason about what the full Llama2-70B model architecture looks like and explain gaps in the simulator data and accuracy, without the data, user, or prompt providing this information. Answering these kinds of questions would greatly aid simulator accuracy validation, which often requires a deep understanding of the model execution.

We describe questions as "valid" if the data contains the answer to the question either by direct lookup or inference. For "valid" questions, we denote the percentage of correct answers provided for all the questions of that type asked. In our evaluation, the questions in each question type were all answered correctly (100%) or incorrectly (0%).

We describe questions as "invalid" if the data does not contain enough information to reason about the answer. For "invalid" questions, a correct answer indicates that the workflow clearly communicated to the user that the answer is not known. In contrast, an incorrect answer means that the workflow did not realize that the answer could not be determined, and worse,

hallucinated an *incorrect* answer. Following the principles of LLM grounding [18], we include invalid questions in our evaluation to stress the importance of not misleading the user when the answer is not known. This is especially important for non-experts who may not have the intuition to realize that the answer is made up.

## V. RESULTS

The workflows are described by whether LLMs or humans are in the loop for the three tasks in the co-design workflow denoted in Figure 1: selecting what data to focus on and use to answer a question, formulating the questions themselves, and reasoning about the answers to these questions. At this time, all workflows require a human to formulate questions. The baseline workflow is Workflow 1, the traditional workflow where only humans are in the loop and all questions are correctly answered. We compare Workflow 1 to workflows with an LLM integrated—either an LLM out of the box, with no prompting (denoted "Gemini 2.0 Flash") or an LLM within the VIEW system, with custom prompting (denoted "VIEW"). For fair comparison, both Gemini 2.0 Flash and VIEW are provided the same pre-processed data before the iterative question-and-answer process.

Workflows 2 and 3 use Gemini 2.0 Flash, out of the box, to generate answers. In Workflow 2, the LLM is not given any direction on what subset of the data to look at. In this case, the model returns incorrect answers. However, the data in these incorrect answers consist of values that exist in the data provided, and when asking Gemini 2.0 Flash to explain its reasoning, its thought process is correct. This behavior indicates issues with potentially exceeding the sequence length for the LLM, suggesting that some understanding of pulling in only necessary subsets of data is necessary for functionality. In Workflow 3, a human provides only the subset of the data relevant for the questions. Then, we find that Gemini 2.0 Flash accurately generates answers for all valid questions, including "Data Query +", without any custom prompting.

However, for invalid questions, we observe that Gemini 2.0 Flash out of the box returns incorrect answers, without any indication that the answer is speculation or a hallucination. For Stage 1 questions, it may not be obvious to experts and non-experts alike that the answer is not correct, since these questions are often asking what the data looks like, rather requiring complex reasoning about consequences. This behavior *lowers* productivity compared to Workflow 1 since the user has to realize that there is an error (if they realize). For less seasoned users, these kinds of answers may result building incorrect intuitions about model and hardware behavior.

Workflows 4 and 5 are tested with the VIEW system. To generate answers with VIEW, we use Gemini 2.0 Flash, with the custom prompting described in Section III-D. The example questions in the prompt are a subset of the questions we ask to evaluate the correctness of the workflow. Importantly, the VIEW is tested on questions that it has not been provided before or given guidance on how to reason about in the prompt.

| Workflow | | | | Percentage of correct answers | | | | |
|---|---|---|---|---|---|---|---|---|
| | | | | Data Query | | Data Comparison | | Data Query + |
| Label | Select Data | Formulate Questions | Generate Answers | Valid | Invalid | Valid | Invalid | Valid |
| 1 | Human | Human | Human | 100% | 100% | 100% | 100% | 100% |
| 2 | None | Human | Gemini 2.0 Flash | 0% | 0% | 0% | 0% | 0% |
| 3 | Human | Human | Gemini 2.0 Flash | 100% | 0% | 100% | 0% | 100% |
| 4 | Human | Human | VIEW | 100% | 100% | 100% | 100% | 100% |
| 5 | VIEW | Human | VIEW | 100% | 100% | 100% | 100% | 100% |

Fig. 3: We evaluate various co-design workflows in answering data queries and data comparisons, with LLMs without prompting ("Gemini 2.0 Flash") and with prompting ("VIEW"). Only Workflows 4 and 5 with VIEW answer all questions correctly (green) like Workflow 1, the traditional workflow, which solely relies on an human expert (purple). Workflow 5, the VIEW prototype in Figure 2, also saves the human time, effort, and expertise required for selecting data and reasoning about answers.

We observe that Workflow 4 answers all the question types correctly. For valid question types, we observed some dangers with overprompting when there are implicit assumptions. Our prompting originally included the below phrasing:

> If the user asks for anything more specific, like what a particular operator is bound by, you can inspect the corresponding row in the dataframe and roofline.

In this example, if we asked the VIEW system whether a layer was compute-bound, memory-bound, or communication-bound, we found that the system found the right compute, memory, and execution times for that layer. In other words, data queries were functional. However, the answer to this boundedness question was often wrong (e.g., returning memory-bound when compute-bound). To understand this behavior, we realized that a result in the "roofline" column contains a single execution time that had previously calculated max(time_compute, time_mem, time_comm). However, which of the three times had been chosen for the roofline was not directly exposed to the LLM in the roofline number in the data. Thus, this prompting directing the LLM to look at a roofline number lead to confusion with this hidden maximum comparison. We instead wanted the LLM to compare these execution times instead of looking up the roofline result, so we removed the above excerpt from the prompt. With this deletion, we observed that the LLM figured out that it needed to compare these three times on its own, and it did not need specific guidance. The LLM then got the right answer, and used these comparisons to explain its reasoning.

Importantly, for invalid questions, VIEW clearly states that the provided data does not contain the information to determine an answer. For example, when asked about the collectives required for Llama2-70B decode, without providing this information in the performance data, VIEW responds that "The provided data does not contain enough information to definitely determine which collectives were required for each layer. The *operand_a*, *operand_b*, and *output* columns show tensor shapes and sharding information but do not explicitly state which collectives were used. More information is needed to answer this question." This behavior can directly be attributed to the part of the prompt shown below:

> When you are pulling knowledge that is not in the data, please explicitly call it out. If in doubt, we prefer that you say that you do not know, rather than provide an incorrect response.

Thus, while LLMs out of the box are valuable for answering valid Stage 1 questions, these invalid questions show the value of custom prompting for an LLM-integrated workflow.

Finally, consider Workflow 5, which corresponds to the VIEW system shown in Figure 2. VIEW has a second call to Gemini 2.0 Flash with a different prompt asking the model to classify types of user questions. The output of this LLM then triggers retrieving different subsets of the data. With this classifier, we can automate the process of deciding what data to pull in, and remove humans from the loop in selecting data. With the same prompting as Workflow 4 for the answer generation, VIEW still maintains correct answers for all questions types, indicating that the data required to answer the question was correctly selected.

## VI. CONCLUSION AND FUTURE WORK

We believe that integrating LLMs to generate *insights* in a hardware-software co-design workflow is a promising direction of research to close productivity and accessibility gaps in the current co-design process. To that end, we reason about progressively complicated stages of questions that users may ask to analyze performance data, building the foundation for developing a benchmark for evaluating LLM-integrated co-design workflows. We demonstrate that our prototype system can successfully automate and speed up part of this workflow, while explaining *how* it arrives to its conclusions.

There are many interesting directions for future work on evaluating LLM-integrated workflows on further stages of questions. While VIEW currently only takes raw performance numbers as input, we could also provide more varied inputs such as simulator code, compiler code, hardware system specifications, and model specifications. We also highlight that improving the "Auto-Select Data" LLM remains an important consideration since providing all of these data inputs can easily exceed the sequence lengths of these models. We could also explore fine-tuning an LLM, in addition to custom prompting and data pre-processing, for targeted results. The LLM could be further augmented with specialized performance analysis tools to automate tasks such as sensitivity analyses.

Providing more data and adding more techniques in this workflow may improve the reasoning capabilities of LLMs.

However, a key challenge is not overcomplicating the solution: we want to determine the minimum viable combinations of data and techniques required for a functional approach. To that end, building a standardized benchmark to evaluate the success rate and cost of LLM-integrated workflows is crucial. We hope this work spurs a discussion on these open questions and encourages further work in this area.

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
