# OpenReview forum: "Leveraging LLMs to Improve Hardware-Software Co-Design Workflow Productivity and Accessibility"
_iscaconf.org/ISCA/2025/Workshop/MLArchSys — MLArchSys 2025 Poster_

### Official Review · Reviewer_F3pg · 2025-05-20
**Leveraging LLMs to Improve Hardware-Software Co-Design Workflow Productivity and Accessibility**

**Confidence:** 4
**Rating:** 3

**Detailed Feedback And Questions For Authors:**

Dear authors, thank you for submitting your work to ML Arch sys 2025.
The submitted manuscript performs a decent job in capturing the idea of the project. As you highlighted in the paper, a tool such as view will certainly democratize the expertize of custom architecture design and recude the turn around time of data analysis and reasoning. However the paper and the high level idea requires a careful scrutiny to highlight its intellectual merit. In my honest opinion as you advance in your project please emphasize on the following point:
1. Actual value proposition of the VIEW tool. While making design decision it is often data generation rather than data analysis which consumes energy and resources. Please ponder if VIEW can solve reduce the cost of the latter
2. Intellectual contribution over state of the art. Contemporary methodologies like RAG, NeuroSymbolic architectures, and novel prompt engineering can achieve refinement of output prompt similar to VIEW's objectives. Please highlight how VIEW can be differentiated against those.
3. Veracity of the tool: As you have mentioned in the paper, it is important to recognize and guard against model hallucination. Please highlight the intellectual contributions and results of such safeguards
4. Evaluation and effectiveness : Please highlight and describe a robust evaluation strategy and benchmark the obtained results against well recognized verifiable benchmarks

**Top Reasons To Accept The Paper:**

+ The paper proposes an idea of using a chat bot assistant to parse raw logs of simulation data and reason about pareto frontiers from the data
+ The tool as described in the paper has potential to democratize the expertize of architecture development

**Top Reasons To Reject The Paper:**

- The paper in the current form depicts only a rough outline for the idea of the tool but does not provide any quantitative information
- There are several important details needed like:
  * How the model has been seleceted?
  * How the raw data is consumed?
  * Information about training/finetuning the LLMs used in VIEW framework?
  * Evaluation methodology
  * Benchmark
  etc

---

### Official Review · Reviewer_xyMc · 2025-05-20
**This paper proposed an LLM-based workflow to facilitate the hardware-software co-design of efficient AI systems.**

**Confidence:** 4
**Rating:** 4

**Detailed Feedback And Questions For Authors:**

In addition to the top reasons for rejection:

Figure 1 is not capturing the traditional co-design problem very well because of the over abstraction: data, questions, etc. It lacks important co-design dimensions such as the optimization targets/constraints in terms of performance, accuracy, power, area, etc. Table-1 adds a bit more clarity with specific examples of questions/answers, but still not an accurate formulation of the hw/sw codesign problem.

The paper writing is a bit redundant which could be improved.

**Top Reasons To Accept The Paper:**

The paper has in-depth analysis on traditional hw/sw co-design process and categorized the q&a and reasoning process into several stages.
The paper identifies the stages that can be facilitated by LLMs: selecting data and generating answers, while still letting human formulate questions.
The paper demonstrates that VIEW with customized prompt engineering can provide better answers to data queries and comparisons.

**Top Reasons To Reject The Paper:**

The paper has assumed a lot of collected performance data and detailed hardware specs, while in reality, the simulation data required for the entire co-design space is enormous and it's usually an iterative process to generate more data and explore deeper into the co-design space. It'd be great if the authors could 1) specify the requirements on the data collection (what are the required data) 2) formulate the workflow as an iterative process of q&a, reasoning and data generation.

The evaluation section uses the correct answers as the metric to evaluate the LLM-based systems. However, correct answers are not the end goals. It'd be better to analyze the co-design decisions: if they are more reasonable, leading to more efficient system designs.

---

### Official Review · Reviewer_4c1q · 2025-05-20
**Leveraging LLMs to Improve Hardware-Software Co-Design Workflow Productivity and Accessibility**

**Confidence:** 3
**Rating:** 5

**Detailed Feedback And Questions For Authors:**

This paper could be strengthened with more technical details and more robust evaluation results. Please consider the following suggestions:

1. Expand the evaluation table to include more information, such as the specific questions used for evaluation. Additionally, create a benchmark set of questions tailored for evaluating such systems. This could serve as an additional contribution and can make evaluation more robust.

2. Incorporate additional evaluation metrics such as scoring the LLM responses by leveraging several state-of-the-art benchmarks to compare the response quality.

3. Break down the evaluation based on different use cases. For example, one use case could involve comparing different hardware architectures such as GPUs and state-of-the-art accelerators. Another use case could involve evaluating different models on the same architecture. This approach would help test whether the LLM prompts are effective across different scenarios.

4. In the evaluation section, you mention the dangers of overprompting. Does VIEW handle this automatically, or does it require user intervention?

5. Integrate actual responses from VIEW rather than only showing example outputs in Table 1 and Figure 2, as this would provide stronger support for your claims.

6. The main contribution of the paper is prompt engineering, but the paper lacks detail on how the prompts were selected. Did you compare different prompts? If so, how did you choose the best one? Do you have a scoring or selection methodology?

7. LLMs typically have a limited context window and tend to lose earlier context once that window is exceeded. How does VIEW address or mitigate this limitation?

8. Include a limitations section to clearly communicate the current constraints of the system.

9. For future directions, as you briefly mentioned, it would be more interesting and innovative to explore training or fine-tuning an LLM that incorporates domain-specific knowledge rather than relying solely on a pre-trained general-purpose model.

**Top Reasons To Accept The Paper:**

1. The paper presents a system that integrates LLMs into the hardware-software co-design workflow, aiming to streamline the process primarily through prompt engineering.

2. The paper raises a key question about leveraging LLMs to enhance co-design efficiency, particularly for non-experts, and highlights some of the potential challenges involved.

**Top Reasons To Reject The Paper:**

1. The paper provides a weak evaluation, with issues such as a lack of detail in the evaluation table about the types of questions evaluated. More details are included below.

2. The paper lacks key technical details, such as how prompts were selected and how they provide sufficient context across different architectures.

---

### Official Review · Reviewer_CfwB · 2025-05-20
**Leveraging LLMs to Improve Hardware-Software Co-Design Workflow Productivity and Accessibility**

**Confidence:** 5
**Rating:** 5

**Detailed Feedback And Questions For Authors:**

Hello,

Thanks for submitting your work. Following is my feedback/comments on your paper:

1. Great work with a prospect of becoming a cornerstone of future research

2. The "Introduction" and "Reasoning about performance data" are well written. They help the readers realize the importance of the probem

3. As much as I like the first two sections, I must mention that for a workshop paper they are bit long. I beleive more focus could be given to actual implementation, more experiments and further evaluation discussions.

4. VIEW provides a UI for the users and in the backend Gemini runs. This hinders the readers to understand the exact novelty of VIEW? Is the UI a novel contribution or incorporating the Gemini ? If first, I don't believe making a nice UI should be considered a novelty. If second, I was expecting tuning of Gemini to allow it to cover diverese cases. The authors should more clearly highlight the contributions of their work (Use bold text/bullets/highlihgts)

5. The choice of Gemini 2.0 Flash is unclear. Why did authros choose this model? Why not others? Is VIEW compatible with other models ? What is the vitality of Custom Prompt and LLM for Auto-Select Data if a newer model supports longer seqeunce lengths? At the same note, what is the vitality of Custom Prompt and LLM for Auto-Select Data if the file size does not exceed the sequenece length of an LLM? Wouldn't it be redundant to keep it there?

6. Prompt engineering does look like a true novelty of this work. But only this can not portray the work as substanially novel.

7. The evaluation is weak. The authors does not discuss any prior works. The authors has to do a thorough evaluation with different LLMs and existing solutions. The effects of hyper-parameters of LLM and how a user can set them ideally must also be studied.

8. Have authors thought about a feedback loop where the answers are generated by LLM and they are fed back again as questions? This way regressively improving the quality of responses ? If it shows improvments and  authors are able to show this in their work, it will be a huge novelty.

Thanks

**Top Reasons To Accept The Paper:**

The paper tackles an important challenge of leveraging LLMs for HW/SW co-design. The introduction is well written and the problem is setup really well in the first two sections.

**Top Reasons To Reject The Paper:**

Lack of evaluation with prior works
Lack of fine tuning the model LLM model to give better results. (VIEW relies complelety on prompt engineering and the performance of LLM itself. What if the LLM model is not good for a problem?)
VIEW compatiblity with Gemini 2.0 Flash only? The authors did not show the evaluation with another LLM.

---

### Official Review · Reviewer_pMbp · 2025-05-20
**The paper aims to formalize the thought-process of hardware architecture and design space exploration with a taxonomy, and automate the process using LLMs. However, the taxonomy and the accuracy metrics are unclear, and there is no comparison with standard chain-of-thought methodologies. I think it is a worthy goal to solve, but the authors should refine the metrics, taxonomy, as well as the evaluation, further.**

**Confidence:** 3
**Rating:** 4

**Detailed Feedback And Questions For Authors:**

Additional questions:
Is there any better error metric than a binary values? Is there a reason to not use standard precision or recall metrics for this work?
For taxonomy, instead of iterating through different phases of questions, would it be more valuable to provide entire series of questions as prompts to the model and let it generalize?

**Top Reasons To Accept The Paper:**

The authors propose using LLMs to automate hardware design space exploration to reduce the human effort as well as turnaround time of the design process. The paper attempts at formalizing the thought process of designing hardware architecture, and proposes a taxonomy for different stages of "questions" posed by an architect. Automating the search of a vast design space is useful and there is value in quantifying the effectiveness/accuracy of generated results for evaluating multiple models for such a complicated task, and there is value in pursuing this problem.

**Top Reasons To Reject The Paper:**

There is not sufficient evidence that the stages for questions specified by the authors are necessarily the kind followed by every architect or user. There is no evidence that the provided taxonomy is generalizable. There is also no comparison with previous work on chain of thought prompting, which could reasonably encompass multiple stage-questions in a single one.
Further, the evaluation and accuracy metrics are unclear. In particular, the correctness is measured in binary (0% or 100%), which does not convey meaningful information. It is unclear how the correctness/error cascades through the different stages, and if chain-of-thought prompting would have provided enough context for better responses by the model instead of single-stage questions.

---

### Official Review · Reviewer_PEaq · 2025-05-20
**This paper proposes a VIEW system, which seamlessly embeds a large language model into the hardware-software co-design process through a "dual LLM" architecture (Auto-Select Data LLM + Generate Answer LLM) and customized prompt engineering. The difficulty of designers' questions about performance data is defined by a four-stage question classification (Stage 1-4) and then tested on a simulation data set. The results show that the accuracy significantly exceeds the Gemini 2.0 Flash basic method without prompts or manual assistance.**

**Confidence:** 3
**Rating:** 6

**Detailed Feedback And Questions For Authors:**

1. It is recommended to supplement the quantitative evaluation or small-scale user research of Stage 2 (causal reasoning), Stage 3 (optimization suggestions), and Stage 4 (meta-question generation) to verify the effectiveness of the system in more difficult tasks.
2. At the data level, it is recommended to consider actual hardware operation logs, etc. to enhance the versatility of the method
3. For the current strict refusal to answer constraints, can it be considered to dynamically adjust at different levels of questions?

**Top Reasons To Accept The Paper:**

- For the first time, LLM is deeply integrated into the hardware-software co-design scenario, and a reasonable problem stratification and evaluation method is proposed
- Aiming at the complexity of the hardware-software co-design scenario, an Agent task flow is designed: Auto-select data, data filter, and generate an answer.  The problem of too long data is avoided through the dual LLM method

**Top Reasons To Reject The Paper:**

Strict "refuse to answer" constraints + lack of data standards and data composition; will strictly limit the applicable scenarios and performance of VIEW.
- Only Stage 1 (data query and comparison) is empirically demonstrated, and there is a lack of quantitative or user research on causal reasoning, optimization suggestions, and meta-question generation (Stage 2-4).

---

### Official Review · Reviewer_EQ6L · 2025-05-21
**This work presents a prototype system, called VIEW, to integrate LLMs for HW/SW Co-Design.**

**Confidence:** 3
**Rating:** 5

**Detailed Feedback And Questions For Authors:**

This work is interesting and timely. It presents a prototype system to integrate LLMs in helping the HW/SW co-design process. The descriptions are written in detail. I think that this paper provides quite useful knowledge/insights for reasoning about data selection and custom prompt engineering for using LLMs for HW/SW co-designs. This can possibly benefit other performance engineering processes too.

When I was reading the paper, however, I felt that the introduction and some earlier parts were describing a bit too broadly than what was actually done in the implementation/evaluation. As an example, data processing from many different types of sources is discussed importantly, but it looks that the implementation/evaluation were based on somewhat simpler case data collected from some simulation; and simulation method for collecting performance data was however a bit too abstracted. Also, some parts of the introduction section could be misleading. For example, when I was reading the introduction, I thought it was targeting general HW/SW co-design problems, but it seems it focuses on HW/SW co-design mostly for DNN chips. Also, in introduction (Fig 1), I thought it was going to integrate LLMs to “formulate questions” , but it still remained as a human’s part.

By the way, this paper is 7 pages long excluding the references and quite longer than the recommended page count (4 pages), although it was not a strict limit.

**Top Reasons To Accept The Paper:**

LLM-assisted HW/SW co-design process is an interesting and timely research problem and matches well with the scope of the workshop. This paper provides quite useful knowledge/insights for reasoning about data selection and custom prompt engineering for using LLMs for HW/SW co-designs.

**Top Reasons To Reject The Paper:**

It was a bit vague to catch the major technical ideas/contributions of the paper, probably due to the structure of the paper.